# Seropositivity-Dependent Association between LINE-1 Methylation and Response to Methotrexate Therapy in Early Rheumatoid Arthritis Patients

**DOI:** 10.3390/genes13112012

**Published:** 2022-11-02

**Authors:** Amin Ravaei, Lia Pulsatelli, Elisa Assirelli, Riccardo Meliconi, Jacopo Ciaffi, Elisa Gremese, Barbara Tolusso, Carlo Salvarani, Marcello Govoni, Michele Rubini

**Affiliations:** 1Medical Genetics Laboratory, Department of Neuroscience and Rehabilitation, University of Ferrara, 44121 Ferrara, Italy; 2Laboratory of Immunorheumatology and Tissue Regeneration, IRCCS Istituto Ortopedico Rizzoli, 40136 Bologna, Italy; 3Medicine and Rheumatology Unit, IRCCS Istituto Ortopedico Rizzoli, 40136 Bologna, Italy; 4Division of Clinical Immunology, Fondazione Policlinico Universitario A. Gemelli IRCCS, 00168 Rome, Italy; 5Division of Rheumatology, Azienda USL-IRCCS di Reggio Emilia, 42122 Reggio Emilia, Italy; 6University Hospital of Modena, University of Modena and Reggio Emilia, 41124 Modena, Italy; 7Section of Hematology and Rheumatology, Department of Medical Sciences, University of Ferrara, 44121 Ferrara, Italy; 8Rheumatology Unit, Sant’Anna University Hospital, 44122 Ferrara, Italy; 9University Center for Studies on Gender Medicine, University of Ferrara, 44121 Ferrara, Italy

**Keywords:** rheumatoid arthritis, LINE-1, DNA methylation, methotrexate, biomarkers, rheumatoid factor, anti-citrullinated protein antibody, pharmacoepigenetics

## Abstract

Background: Methotrexate (MTX) is considered the first choice among disease-modifying anti-rheumatic drugs (DMARDs) for rheumatoid arthritis (RA) treatment. However, response to it varies as approximately 40% of the patients do not respond and would lose the most effective period of treatment time. Therefore, having a predictive biomarker before starting MTX treatment is of utmost importance. Methylation of long interspersed nucleotide element-1 (LINE-1) is generally considered a surrogate marker for global genomic methylation, which has been reported to associate with disease activity after MTX therapy. Methods: We performed a prospective study on 273 naïve early RA (ERA) patients who were treated with MTX, followed up to 12 months, and classified according to their therapy response. The baseline LINE-1 methylation levels in peripheral blood mononuclear cells (PBMC) of cases were assessed by bisulfite pyrosequencing. Results: Baseline LINE-1 methylation level per se turned out not to predict the response to the therapy, nor did age, sex, body mass index, or smoking status. However, if cases were stratified according to positivity to rheumatoid factor (RF) and anti-citrullinated protein antibody (ACPA) or seronegativity, we observed an opposite association between baseline LINE-1 methylation levels and optimal response to MTX therapy among responders. The best response to MTX therapy was associated with hypermethylated LINE-1 among double-positive ERA cases (*p*-value: 0.002) and with hypomethylated LINE-1 in seronegative ERA patients (*p*-value: 0.01). Conclusion: The LINE-1 methylation level in PBMCs of naïve ERA cases associates with the degree of response to MTX therapy in an opposite way depending on the presence of RF and ACPA antibodies. Our results suggest LINE-1 methylation level as a new epigenetic biomarker for predicting the degree of response to MTX in both double-positive and seronegative ERA patients.

## 1. Introduction

Rheumatoid arthritis (RA) is an inflammatory autoimmune disease characterized by progressive bone, cartilage, and joint destruction [1] that affects approximately 0.5–1% of the worldwide population [2]. RA aetiology is still mainly unknown but there is general agreement on considering it as a multifactorial condition due to the interplay between genetic factors [3] and environmental exposure [4,5], and characterized by specific epigenetic modifications [6,7,8,9]. Early RA (ERA) is denoted to patients with a disease duration of fewer than 2 years, but currently most rheumatologists preferentially refer to cases with onset of symptoms of less than 6 months [10].

According to the recommendation of the European League Against Rheumatism (EULAR) and the American College of Rheumatology (ACR), methotrexate (MTX) is the first-line treatment for RA [2]. However, around 40% of patients do not sufficiently respond to MTX [11] within the period that is known as the “window of opportunity” after which treatment becomes less effective [12], resulting in prolonged inflammation and subsequent irreversible joint damage and functional disability [13]. A 3 to 6 months period of “trial and error” is still the main treatment approach due to unknown MTX responses [13] and therefore there is an imperative need for response-predicting biomarkers for early identification of MTX-unresponsive patients, to promptly divert them to other therapies.

DNA methylation, which is the most well-studied, stable, and easily comparable epigenetic alteration [14,15], mainly occurs by transferring a methyl group to the fifth carbon position of cytosine at cytosine–phosphate–guanine dinucleotides (CpG) by DNA methyltransferases (DNMTs) [16]. CpGs are rare across the genome and are mainly methylated [14], clustered in the promoter region of genes, called CpG islands, and usually hypomethylated in transcriptionally active genes [17]. As the DNA methylation signature of some CpGs has previously been correlated with RA onset [6,7,8,9,18,19,20,21,22,23,24,25,26], recent studies explored the effect of MTX treatment on DNA methylation changes [20,27,28] or associated DNA methylation signature with response to MTX therapy [29,30,31,32].

Differences in baseline genome-wide DNA methylation, known as global methylation, have been reported to associate with disease activity after MTX therapy [32]. Approximately 55% of the human genome was constituted by repetitive elements [33,34], among which long interspersed nucleotide element-1 (LINE-1) by covering about one-sixth of the human genome [35,36] is considered a highly repetitive interspersed element. Therefore, LINE-1 methylation is generally considered a surrogate marker for global genomic methylation [37] and its methylation changes could be associated with response to MTX therapy in RA patients.

We aimed to investigate the association of LINE-1 methylation pattern of early rheumatoid arthritis (ERA) patients at baseline with the outcome of their treatment with MTX.

## 2. Materials and Methods

A total of 364 patients with early RA (ERA) were recruited from 2013 to 2016 at the Rheumatology units of four Italian hospitals, including the Sant’Anna University Hospital, Ferrara; the Rizzoli Orthopaedic Institute, Bologna; the S. Maria Nuova Hospital, Reggio Emilia; and the Policlinico A. Gemelli-Catholic University of the Sacred Heart, Roma. All ERA patients included in the investigation were diagnosed according to the ACR and EULAR criteria of 2010 [1] and provided written informed consent to participate in the study. A peripheral blood sample was collected from each case at baseline, prior to administration of 7.5–15.0 mg of oral MTX. Patients were followed up for 12 months, and response to the therapy was evaluated according to EULAR response criteria [38,39] and categorized as good response (GR), moderate response (MR), and no response (NR) (Table 1). The study was reviewed and approved by the ethical board of the University of Ferrara (N. 01/2013). For each case included in the study, baseline serological data were collected, including rheumatoid factor (RF) and anti-citrullinated protein antibodies (APCA), along with sex, body mass index (BMI), and current exposure to tobacco smoking.

Genomic DNA was extracted from peripheral blood samples using QIAamp DNA Blood Mini (Qiagen, Leipzig, Germany) or Nucleon BACC1 (GE Healthcare, Hatfield, UK) kits according to the manufacturer’s instructions. DNA was quantified by a Qubit 2.0 fluorometer (Invitrogen, Singapore) using Qubit dsDNA BR assay kit (Life Technologies, Milwaukee, WI, USA). The gDNA was bisulfite converted using the EZ DNA Methylation-Lightning Kit (Zymo Research, Irvine, CA, USA) according to the manufacturer’s instructions. LINE-1 promoter amplification and assessment of its methylation were carried out as previously reported [40].

The association of age, sex, tobacco smoking, and serological characteristics of the patients with the response to the MTX was evaluated by chi-square test (OR) and reported as odds ratio (OR) with a 95% confidence interval (CI). The average methylation rates at five CpG sites in LINE-1 promoter were compared among case groups using the unpaired Welch’s t-test and results were reported as mean ± SEM. Computations were first performed to detect the association between LINE-1 methylation and response to MTX therapy by comparing responders (GR and MR groups) with the NR group. Next, the association between LINE-1 methylation and optimal response among responders was checked by comparing GR cases with MR cases. Stratification of cases according to age class, sex, BMI, exposure to tobacco smoking, and positivity to RF and/or APCA was included in computations. In particular, regarding age, cases were classified considering the age of 60 as the threshold. Regarding BMI, cases were classified using a threshold of 25 kg/m^2^. Statistical significance was defined at the 95% level (*p* = 0.05) and adjusted according to Bonferroni corrections for multiple comparisons when required. The odds ratio (OR) was calculated to evaluate the influence of specific parameters on therapy response. All computations were performed using Microsoft Excel (2016).

## 3. Results

### 3.1. Patient Characteristics and Therapy Response

Out of 364 recruited patients, 273 were included in the analyses after quality control of data. Details are presented in Table 2. As expected, around three-quarters of the patients were female, with an average age of 59.2 ± 16.0 years, while male patients were on average 650.2 ± 13.1 years old. Overall, approximately half of the cases included in the study were over 60 years old. At the time of ERA diagnosis, the disease length of patients was on average 16.9 ± 13.4 weeks, and the DAS28 score was on average 5.05 ± 1.30. BMI of cases was on average 26.1 ± 5.2 kg/m^2^, and approximately half of them presented with BMI > 25 kg/m^2^. Among this group, two-thirds (66.9%) were overweight and one-third (33.1%) were obese (>30 kg/m^2^). As many as 44.7% of the patients reported being currently exposed to tobacco smoking. Regarding serological characteristics, 56.4% of cases were positive for either RF or ACPA or both (seropositive patients), while 40.3% were positive for both.

After the 12-month follow-up period, one patient out of seven (13.9%) resulted with no response to the MTX therapy. Among the responder group, one-third (33.6%) were classified as MR, and two third (66.4%) showed a good response (GR) to the therapy (Table 2).

Among the studied parameters, only the exposure to tobacco smoking turned out significantly associated with response to the therapy (GR + MR vs. NR) (OR = 2.39, 95% CI [1.18–4.87], *p* = 0.015), but after adjusting for multiple comparisons it turned out not significant (*p* = 0.057). Among MTX responders, the male patients showed a trend towards association with good response to therapy compared to females (OR = 2.002, 95% CI [1.023–3.917], *p* = 0.042). No association between serological parameters and response to therapy was observed.

### 3.2. Association with LINE-1 Methylation Level

Among the 273 studied cases, the LINE-1 methylation level at baseline was 67.27% on average. As shown in Figure 1, no significant differences were observed among the three response groups. The mean methylation level of non-responders was 67.03 ± 0.71% while it was 67.31 ± 0.22% among MTX responders. Amidst these latter cases, the mean methylation of good responders was 67.24 ± 0.29%, and 67.44 ± 0.36% among moderate responders.

The correlation of LINE-1 methylation level with response to the therapy was further assessed considering the sex, age class, BMI class, or smoking status of the patients. As presented in Table 3, the LINE-1 methylation level in female or male patients was very similar across all three therapy response groups, and stratification by sex turned out to show no association between LINE-1 methylation and response to therapy. Similar results were obtained after stratification by age class (threshold at age of 60 years) or by exposure to tobacco smoking.

The stratification of cases based on RF positivity resulted in no effect on the association between baseline LINE-1 methylation level and no response to MTX therapy (NR vs. responders). However, splitting responders in RF positive (RF+) and RF negative (RF−) subgroups resulted in a nominally significant association with optimal response to therapy (GR vs. MR), but with opposite orientation (Figure 2). While hypermethylation of LINE-1 was associated with optimal response to therapy among RF+ cases (nominal *p*-value: 0.020), it was inversely associated among the RF− patients (nominal *p*-value: 0.016), although the comparisons did not reach a significant level after Bonferroni correction (corrected *p*-values: 0.114 for RF+ and 0.092 for RF− cases).

The analysis based on ACPA status showed results very similar to those obtained by RF stratification (Figure 3). While stratification by ACPA-positivity resulted in no association between LINE-1 methylation level and no-response to MTX-therapy, in the ACPA-positive (ACPA+) stratum of MTX-responders the LINE-1 methylation level was positively associated with good response to therapy (nominal *p*-value: 0.016), while it was inversely associated in the ACPA-negative (ACPA−) stratum (nominal *p*-value: 0.011). However, differences between LINE-methylation in the MR and GR groups turned non-statistically significant after Bonferroni correction (corrected *p*-values: 0.092 for ACPA+ and 0.064 for ACPA− cases).

The similar trends towards an association between LINE-1 methylation and high response to MTX therapy obtained after stratification by RF or by ACPA led us to investigate if seropositivity, e.g., the presence of at least one of the two types of antibodies [41], was a better performing predictive biomarker. The stratified analysis, however, did not result in significant improvement as the nominal *p*-value was 0.051 among seropositive cases and 8.53 × 10^−3^ among the seronegative ones. As shown in Figure 4, among the seropositive cases the association between LINE-1 methylation level and rate of response to MTX therapy seems restricted to double-positive cases (RF+ & ACPA+) only (*p*-value: 2.11 × 10^−3^) in which the mean methylation level was 2.06% higher among the GR compared to MR. Instead, no methylation difference between GR and MR was observed among cases with either RF+ or ACPA+.

## 4. Discussion

MTX is the conventional synthetic DMARD of the first choice in RA, but therapy response varies widely among patients, and approximately 40% of them discontinue MTX in the medium term due to inefficacy or adverse events [42]. The “trial and error” approach for MTX therapy inevitably leads to therapeutic failures on part of the patients, which could be avoided if predictive markers with sufficient accuracy would be available to identify non-responsive cases before starting MTX therapy and enable earlier access to alternative medications to avoid disease progression.

Recently, a study led by S.G. Heil provided evidence supporting that higher baseline global DNA methylation in PBMC of naïve ERA patients was associated with a reduced decrease of disease activity score 28 (DAS28) after three months of MTX therapy [32] and suggested this as a predicting biomarker to identify non-responding cases. The authors also performed a validation study in LINE-1 elements focusing on seven GpGs and reported that methylation at only one of them was associated with DAS28 changes after MTX therapy.

Our study focused on methylation at five CpGs in the promoter of LINE-1 elements which are generally considered a good surrogate of global genomic DNA methylation and are located slightly upstream of the sequence investigated in Heil’s study (Figure 5). We evaluated the LINE-1 methylation level in the peripheral blood cells of naïve ERA patients and found no significant association with the response to first-line therapy with MTX. This result is in line with Heil’s study reported in six out of seven CpG sites, which were presenting methylation levels not associated with DAS28 changes [32].

Liebold and colleagues carried out a study aimed to detect the global genomic DNA methylation in RA patients at baseline and after 3 months of MTX therapy in both PBMCs and subtype cells by flow-cytometry measurement of 5-methyl-cytosine levels [30]. Although the study did not directly provide the correlation between baseline global DNA methylation and the outcome of the therapy, it demonstrated that global DNA methylation levels were associated with disease activity and suggested that this biomarker could be used for therapy monitoring and clinical outcome prediction. In particular, the study showed that in active RA patients the DNA from PBMCs or leukocyte subsets was significantly hypomethylated compared to healthy donors and that in patients responding to MTX therapy the global DNA methylation was significantly increased. Recently, the MATURA study performed the first epigenome-wide methylation study (EWAS) aimed to detect differentially methylated positions (DMPs) associated with the response to MTX therapy in RA patients [29]. The study used the Illumina Human Methylation450 BeadChips platform and detected DMPs with no a-priori hypothesis at the genome-wide level by comparing the DNA methylation levels at baseline and after 4 weeks of MTX therapy. The MATURA study led to the identification of 12 independent DMPs corresponding to specific CpGs whose methylation level correlated with changes in specific DAS28 components. The results of the study were validated in four CpGs using pyrosequencing in an independent cohort of RA patients. However, the study could not identify any DMP that at baseline was able to predict the response to MTX therapy, as the baseline DNA methylation level in good and poor responders was not significantly different.

Recently, S.G. Heil’s group conducted another EWAS in PBMCs-derived DNA from ERA patients to identify baseline DMPs and correlate them with MTX response at 3 months of monotherapy [31]. Compared to the MATURA study, Heil’s group study used a different platform—the Illumina Infinium Methylation EPIC platform—but again detected no significant DMPs that at baseline were associated with the response to MTX therapy. Therefore, based on the EWAS using PBMCs in naïve ERA patients, it seems that no specific genomic position presents variations in methylation level (epimutations) that alone can predict the response to the therapy with MTX. This, however, does not exclude that interplay between multiple DMPs, or interactions between DMPs and specific clinical aspects presented at the time of RA diagnosis, could in the future turn out to associate with MTX response. In addition, it is important to take into account that there are hundreds of thousands of copies of LINE-1 sequence in the human genome, and that the average level of LINE-1 methylation is likely an oversimplification of the epigenome. Due to the repetitive nature and multi-mapping issue of short reads, Illumina Infinium Methylation EPIC platform is often not suitable for LINE-1 related studies, while long-reads sequencing platforms, such as the Nanopore WGS, could be more appropriate.

The results of our study showed that parameters such as sex, age, BMI, and smoking status seem not to interplay with LINE-1 methylation levels in PBMCs of naïve ERA cases to associate with their response to MTX therapy. However, when baseline serological characteristics were considered, we found that, among MTX responders, the positivity to RF and ACPA significantly influences the association between LINE-1 and the degree of the response to therapy, showing an opposite effect in double-positive and seronegative cases. As the methylation in LINE-1 has opposite effect depending on the RF/ACPA positivity or negativity, if ERA cases are not stratified in these serological groups the association between LINE-1 methylation level and magnitude of response to therapy is reset. To our knowledge, this is the first evidence that the interaction between RF/ACPA seropositivity and LINE-1 methylation level in PBMCs at baseline associates with the degree of response to MTX therapy.

It is known that RA patients with increased serum levels of RF or ACPA are at higher risk of joint damage and a more severe disease course compared with seronegative patients [43], and in seropositive patients, the related autoantibodies develop years before RA diagnosis [44]. In particular, the combined positivity to RF and ACPA in RA patients was associated with an increased level of pro-inflammatory cytokines and C-reactive protein, which was hypothesized to be a consequence of the amplifying effect of RF on tumour necrosis factor (TNF) production triggered by ACPA [45] whereby ACPA interacts with citrullinated protein to form immune complexes (ICs) which activates the complement system, recruits, and activates immune cells, thereby triggering the production of proinflammatory cytokines [46]. The boosting activity of RF on ACPA background was confirmed in several clinical trials, as single positive (ACPA+, RF-) patients had a lower level of disease activity compared to double-positive ones [47]. Moreover, studies of two large independent ERA cohorts have provided evidence that triple-positive patients—i.e., with RF, ACPA, and anti-carbamylated protein antibodies—had the highest levels of acute phase reactants, suggesting that the amount of inflammation in ERA is proportional to the number of autoantibody-specificities [48].

The concordance of RF+ and ACPA+ in RA is somehow controversial, as observations indicate that ACPA+ patients had disease activity that was similar to, or indeed less than, that of ACPA− patients, both in presence or absence of RF [47]. Moreover, in ACPA+/RF− patients higher ACPA concentrations have been associated with an increased likelihood of remission [49].

The evidence that hypermethylation of LINE-1 associates with good response to MTX among double positive ERA cases inversely associates in seronegative patients, while has no effect in single positive cases, suggests a possible interplay between LINE-1 methylation level and the degree of loss of tolerance to autoantigens, concerning the efficacy of MTX in counteracting the progression of the disease. Moreover, this evidence further supports the notion that the simple distinction between seropositive and seronegative patients is insufficient at capturing the different facets of the disease, and that double-positive and single-positive patients should be considered separately, by taking into account that autoantibodies might have different mechanisms in disease induction. For instance, evidence was reported supporting a novel mechanism of disease induction in which ACPAs act as agonists for receptor-mediated responses that directly promote osteoclastogenesis and induce joint pain [50].

The limited number of cases per MTX-response group may have decreased the statistical power and the number of significant values. This was also partly due to the initial quality control selection of the enrolled patients, a quarter of whom were excluded due to the incompleteness of clinical data or insufficient quality of biological samples.

As this study is based on the analysis of genomic DNA obtained from peripheral blood, the results should be interpreted cautiously by considering that the subtypes of peripheral blood cells could differ in seropositive and seronegative patients. In the next studies, it may be more appropriate to focus on specific cell types.

## 5. Conclusions

Our study provides evidence that in naïve ERA cases the LINE-1 methylation level in PBMCs associates with the degree of response to MTX therapy in an opposite way depending on RF/ACPA double-positivity or seronegativity. This result is to be considered preliminary and objectively not sufficient to develop explanatory hypotheses, nor to identify causal relationships. However, although LINE-1 methylation level alone does not associate with MTX response in ERA patients, it could be used as an epigenetic biomarker for predicting the degree of response to MTX if integrated with RF/ACPA assessment. Further investigations are awaited to replicate our results and eventually pave the way for the combined use of LINE-1 methylation and RF/ACPA assessment for selecting the patients who have the best chance to respond to MTX therapy, in a view of an epigenetic biomarker-based approach of personalized therapy.

## Figures and Tables

**Figure 1 genes-13-02012-f001:**
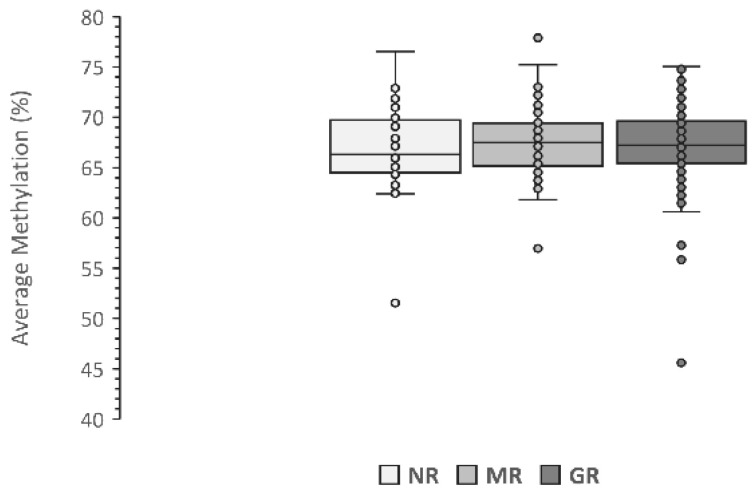
Average LINE-1 methylation level in relation to the response to MTX therapy. Data are presented as the average methylation percentage of five CpG sites. NR, No Response; MR, Moderate Response; GR, Good Response.

**Figure 2 genes-13-02012-f002:**
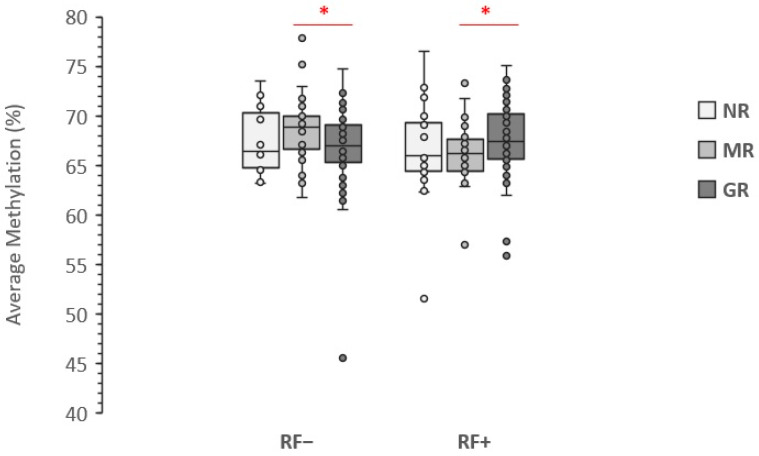
Association of LINE-1 methylation level with response to the therapy stratified by RF positivity. Data are presented as the average methylation percentage of five CpG sites. NR, No Response; MR, Moderate Response; GR, Good Response; * significant nominal *p*-value.

**Figure 3 genes-13-02012-f003:**
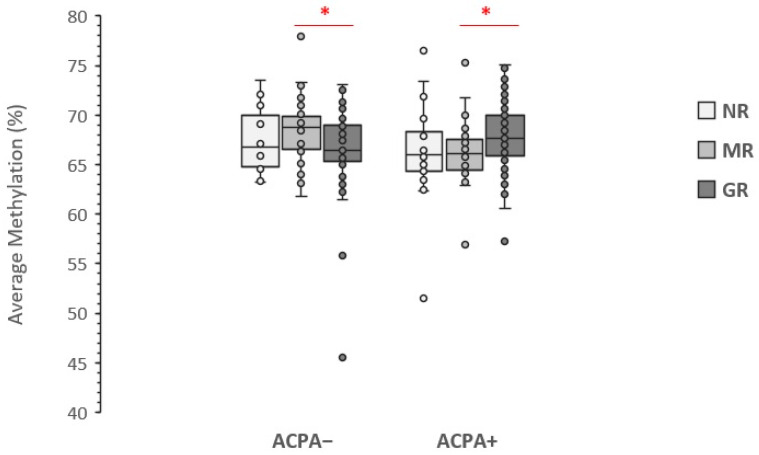
Association of LINE-1 methylation level with response to the therapy stratified by ACPA positivity. Data are presented as the average methylation percentage of five CpG sites. NR, No Response; MR, Moderate Response; GR, Good Response; * significant nominal *p*-value.

**Figure 4 genes-13-02012-f004:**
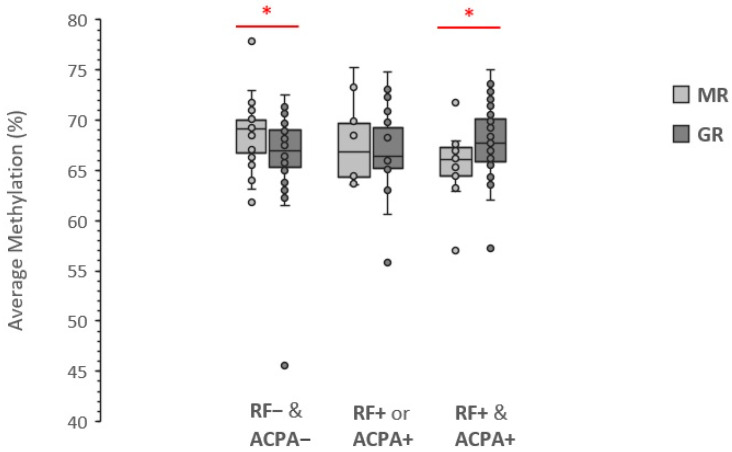
Effect of seropositivity on the association between LINE-1 methylation level and the magnitude of response to MTX therapy. Data are presented as the average methylation percentage of five CpG sites. FR− & ACPA−: seronegative; FR− or ACPA−: single positive; FR+ & ACPA+: double-positive; MR, Moderate Response; GR, Good Response; * significant nominal *p*-value.

**Figure 5 genes-13-02012-f005:**
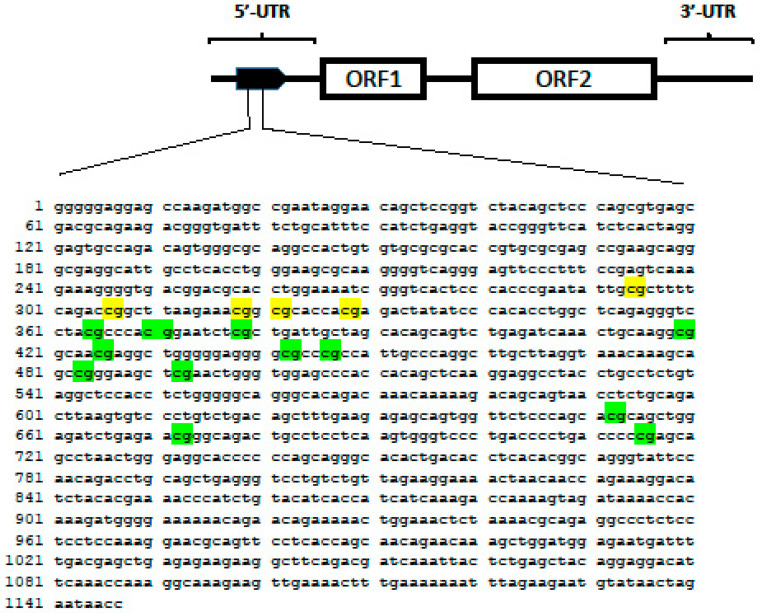
CpG island region of the human LINE-1 transposon (Locus X58075). The five CpG sites highlighted in yellow were investigated in the present study. The twelve CpG sites shown in green were studied by Gosselt et al. [31].

**Table 1 genes-13-02012-t001:** EULAR therapy response criteria using DAS28.

Present DAS28	DAS28 Improvement		
	>1.2	>0.6 and ≤1.2	≤0.6
≤3.2	Good response	Moderate response	No response
>3.2 and ≤5.1	Moderate response	Moderate response	No response
>5.1	Moderate response	No response	No response

**Table 2 genes-13-02012-t002:** Characteristics of the patients included in the study (*n* = 273).

Variable	Stratum	(%)
Sex	Female	73.7
Age	>60 years	51.6
BMI ^1^	>25 kg/m^2^	54.0
Tobacco smoking	Current smokers	44.7
RF ^2^	Positive	49.8
ACPA ^3^	Positive	47.4
RF/ACPA ^4^	Positive	56.4
MTX response	NR ^5^	13.9
	MR ^6^	28.9
	GR ^7^	57.2

^1^ Body mass index; ^2^ Rheumatoid Factor; ^3^ Anti-Citrullinated Protein Antibodies; ^4^ Positive to RF and/or ACPA; ^5^ No Response; ^6^ Moderate Response; ^7^ Good Response.

**Table 3 genes-13-02012-t003:** Average LINE-1 methylation levels with response to the therapy considering sex, age, BMI, and smoking status.

	NR ^1^			MR ^2^			GR ^3^		
	Patients (N)	Average Methylation (%)	SEM	Patients (N)	Average Methylation (%)	SEM	Patients (N)	Average Methylation (%)	SEM
**Sex**									
Female	27	67.40	0.94	65	67.19	0.41	109	67.31	0.34
Male	11	66.11	0.81	14	68.60	0.69	47	67.09	0.53
**Age**									
>60	18	67.24	1.30	43	67.78	0.55	80	67.09	0.44
≤60	20	66.84	0.70	36	67.05	0.45	76	67.40	0.37
**BMI**									
≤25	14	66.29	1.59	31	67.57	0.62	76	67.17	0.37
>25	24	67.46	0.65	44	67.31	0.47	74	67.24	0.45
**Smoking**									
Non-smokers	14	67.24	0.84	51	67.80	0.43	86	67.45	0.41
Smokers	24	66.90	1.02	28	66.79	0.66	70	66.99	0.39

^1^ No Response; ^2^ Moderate Response; ^3^ Good Response.

## Data Availability

The datasets generated during the study are available from the corresponding author upon reasonable request.

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
