# Peer review of "Seropositivity-Dependent Association between LINE-1 Methylation and Response to Methotrexate Therapy in Early Rheumatoid Arthritis Patients"

_genes, 2022, doi:10.3390/genes13112012_

Round 1

Reviewer 1 Report

This intriguing study entitled “Seropositivity-dependent association between LINE-1 methylation and response to methotrexate therapy in Early Rheumatoid Arthritis patients” reports that by stratifying cases of early Rheumatoid Arthritis (ERA) according to positivity to  rheumatoid factor (RF) and anti-citrullinated protein antibody (ACPA) or, in contrast, sero-negativity to both RF and ACPA, they found an opposite (mirror image) association between baseline LINE-1  methylation levels and optimal response to methotrexate (MTX) therapy among responders. 

General points

The authors set out to find a predictive biomarker of response before initiation of MTX therapy, and this is indeed a much sought-after goal in treatment of RA – the findings here are of potential high importance. This prospective study on 273 naïve early RA (ERA) patients who were treated with MTX is carefully planned and well executed. The patients were followed up for 12 months, evaluated according to EULAR response criteria [38,39] and classified according to their therapy response into 3 different categories - good response (GR), moderate response (MR), and no response. The best response to MTX therapy was associated with hypermethylated LINE-1 among double-positive ERA cases (p-value 0.002) & with hypomethylated LINE-1 in seronegative ERA patients (p-value 0.01). 

 The authors propose LINE-1 methylation level as a new epigenetic biomarker for predicting the degree of response to MTX in both double-positive and seronegative ERA patients.

Specific points

 The authors report the importance of seropositivity in patients with early rheumatoid arthritis (ERA) for treatment response to MTX and the combined effects of RF & ACPA mentioned in the context of double positive ERA patients.

It is worth noting that the concordance of RF & ACPA in RA is somewhat controversial. For instance Aletaha et al. in a 2015 article entitled “Rheumatoid factor, not antibodies against citrullinated proteins, is associated with baseline disease activity in rheumatoid arthritis clinical trials” reported that ACPA positive patients had disease activity that was similar to, or indeed less than that of ACPA negative patients, both in presence and absence of RF. https://pubmed.ncbi.nlm.nih.gov/26307354/

In this series, 56.4% of cases were positive for either RF or ACPA or both, while 40.3% were positive for both. These figures are significantly lower that the 70% seropositivity rates (approximate) reported for patients from further north in Europe, and this may reflect a higher incidence of HLA-DR4/DR1 and shared epitope positivity in those populations.

Miriovsky et al. in 2010 also found in ACPApos/RFneg patients that higher ACPA concentrations were associated with an increased likelihood of remission. https://pubmed.ncbi.nlm.nih.gov/20439294/

This current study focuses on the enhancing effect of RF, which tiggers production of pro-inflammatory cytokines such as TNF. This production itself is triggered by ACPA. This mechanism has been previously reported by Cavel et al. in 2008 for ACPA in disease induction, whereby ACPA interacts with citrullinated protein to form immune complexes (IC) which activates the complement system, recruits and activates immune cells, thereby triggering production of proinflammatory cytokines. https://pubmed.ncbi.nlm.nih.gov/18311806/

 Since the present study focuses heavily on autoantigens, it is perhaps worth noting that there is another novel mechanism of ACPA in disease induction (Krishnamurthy A et al., 2016). In this mechanism, ACPA does not bind citrullinated antigen and acts as an agonist for a receptor-mediated response that directly induces pain and osteoclastogenesis.  https://pubmed.ncbi.nlm.nih.gov/26612338/

It is recommended that some of these points might be addressed in the Discussion. 

Typo in Line 187: RF (instead of FR)

Reviewer 2 Report

The author used a previously established LINE-1 promoter methylation assay on 364 patients blood samples.

There seems to be an association between the LINE-1 promoter methylation levels and response levels of MTX treatment.  

I have a few points to share, perhaps they could improve the manuscripts.

  1. The general effect sizes of the difference between MR, NR, GR are very small. even though they are significantly different, p-value ~ 0.002. This association study suggests that differences in treatment outcome is associated with differences in methylation levels. The authors perhaps should clarify that the evidence provided in the paper does not suggest a causal relationship. However, if the casual relationship can be investigated or suggested a mechanism, it could improve the impact of the paper.

  2. Only the average methylation rates at five CpG sites are reported. I am not sure if the detailed information of each CpG site can be provided. If some explanations on the average 17 (5+12) CpG sites can be provided, it will better support the LINE-1 promoter methylation level. Perhaps, please provide an explanation why these five CpG are selected.   

3. There are hundreds of thousands of copies of LINE-1 in the human genome. The average methylation perhaps is an oversimplification of the epigenome. Due to the repetitive nature and multi mapping issue of short reads, Illumina WGBS is often not sustiale for LINE-1 related studies. Nanopore long reads WGS can provide 5mC levels at each individual CpG site. These ideas might be included as future directions in the Discussion section. 
